

# Microbial diversity patterns in the root zone of two *Meconopsis* plants on the Qinghai-Tibet Plateau

Shuting Chen[1],*, Pengxi Cao[1],*, Ting Li[1], Yuyan Wang[1] and Xing Liu[1,2]

[1] Laboratory of Adaptation and Evolution of Plateau Biota to Extreme Environments, School of Ecology and Environment, Tibet University, Lhasa, China

[2] State Key Laboratory of Hybrid Rice, Key Laboratory of Biodiversity and Environment on the Qinghai-Tibet Plateau, Ministry of Education, College of Life Sciences, Wuhan University, Wuhan, China

* These authors contributed equally to this work.

Corresponding author
Xing Liu, xingliu@whu.edu.cn

## ABSTRACT

In the extreme alpine climate of the Qinghai-Tibet Plateau (QTP), plant growth and reproduction are limited by extremely cold temperatures, low soil moisture, and scarce nutrient availability. The root-associated microbiome indirectly promotes plant growth and plays a role in the fitness of plants on the QTP, particularly in Tibetan medicinal plants. Despite the importance of the root-associated microbiome, little is known about the root zone. This study used high-throughput sequencing to investigate two medicinal *Meconopsis* plants, *M. horridula* and *M. integrifolia*, to determine whether habitat or plant identity had a more significant impact on the microbial composition of the roots. The fungal sequences were obtained using ITS-1 and ITS-2, and bacterial sequences were obtained using 16S rRNA. Different microbial patterns were observed in the microbial compositions of fungi and bacteria in the root zones of two *Meconopsis* plants. In contrast to bacteria, which were not significantly impacted by plant identity or habitat, the fungi in the root zone were significantly impacted by plant identity, but not habitat. In addition, the synergistic effect was more significant than the antagonistic effect in the correlation between fungi and bacteria in the root zone soil. The fungal structure was influenced by total nitrogen and pH, whereas the structure of bacterial communities was influenced by soil moisture and organic matter. Plant identity had a greater influence on fungal structure than habitat in two *Meconopsis* plants. The dissimilarity of fungal communities suggests that more attention should be paid to fungi-plant interactions.

# INTRODUCTION

The Qinghai-Tibet Plateau (QTP) is the largest ice storage area on earth, with low oxygen and harsh climatic conditions, and is known as the third pole of the earth (*Qiu, 2008*). Due to its unique geographical characteristics, there is a significant temperature difference between daytime and nighttime on the QTP (*Liu et al., 2013*). Plants on the QTP have evolved the morphological and physiological characteristics necessary to survive under

extreme cold and drought stress (*Sun et al., 2014*; *Boucher et al., 2016*; *Xu et al., 2021*). The root-associated microbiome also plays a role in the fitness of plants, especially in conditions of extreme cold and drought stress (*Acuña-Rodríguez et al., 2020*; *Trivedi et al., 2020*). The root microbiome is created from the surrounding bulk soil (*Yuan et al., 2015*), and can induce changes in the root exudates of plants suffering from poor nutrition, which then affects the structure of the rhizosphere microbiome. Following this pattern, plants produce beneficial microbes to indirectly promote their environmental tolerance, pathogen resistance, and fitness, which is of great significance to agricultural production (*Bulgarelli et al., 2013*; *Brunner et al., 2015*; *Bakker et al., 2018*; *Fitzpatrick et al., 2018*; *Hu et al., 2018*; *Yuan et al., 2018*; *Otto, 2021*). The root microbiome also influences the stabilization of soil organic carbon storage on the QTP over a long timescale, which is particularly important in the context of global warming (*Ruess et al., 2003*; *Moore et al., 2015*; *Tian et al., 2021*).

Bacteria and fungi make up the majority of soil microbial communities (*Fierer, 2017*). Biotic and abiotic factors, such as host genotype and soil type, influence the microbial composition of the roots (*Edwards et al., 2015*; *Agler et al., 2016*). Soil type is the main factor in the pH, temperature, moisture, and nutrient availability of the soil (*Bååth & Anderson, 2003*; *Pietri & Brookes, 2009*; *Li et al., 2016*; *Na et al., 2019*). The composition of the root microbiome is influenced by host-related genotype, growth stage, and microbial competition (*Chaparro, Badri & Vivanco, 2014*; *Castrillo et al., 2017*; *Dombrowski et al., 2017*; *Stringlis et al., 2018*). In *Boechera stricta* (Graham), host genotype had a greater impact on the leaf microbiome than the root microbiome, and the composition of the root microbiome was host-specific and changed with age (*Wagner et al., 2016*). A previous study found that bacterial communities showed significant differences due to geographical location, even among the same plant species (*Yamamoto et al., 2020*). Most research on the effects of plant identity or habitat on microbial communities focuses on endophytes and rhizosphere communities (*Glynou et al., 2016*; *Wippel et al., 2021*; *Zuo et al., 2021*; *Maciá-Vicente & Popa, 2022*), but the impact of plant identity or habitat on the root microbiome has received far less attention. The soil in the root zone is an important component of the soil-root system (*Shi et al., 2019*). This study of the root zone fills in a gap in the current research. A few studies have shown that plant identity and geographical location impact microbial structure. The tree oak clone DF159 was transplanted into various habitats to serve as an environmental measuring instrument, and it was found that environmental variables had a stronger impact than plant identity on the microbial communities of the root zone (*Habiyaremye et al., 2020*). *Bintarti et al. (2020)* reported that the microbial diversity in the root zone of apple trees was strongly influenced by geographical location. It is still unclear, however, whether plant identity or geographical factors have a greater impact on the microbial composition of the root zone of plants on the Qinghai-Tibet Plateau.

*Meconopsis*, of the Papaveraceae family, is mainly distributed in the Himalayas of China, except for one species that grows in Western Europe (*Wu, 1980*). Because of their resistance to cold temperatures and ultraviolet radiation, they grow at high altitudes and in low temperatures. *Meconopsis* is a traditional Tibetan medicinal plant that contains

bioactive compounds such as alkaloids, flavonoids, volatile oils, and triterpenes. Alkaloids and flavonoids are the main secondary metabolites of *Meconopsis* (*Zhang, Li & Zhou, 1997*; *Liu et al., 2014*; *Yun et al., 2015*). *Meconopsis horridula* Hook. f. & Thomson is used for its antitumor properties and *Meconopsis integrifolia* (Maxim.) Franch has hepatoprotective and antioxidant functions (*Zhou et al., 2013*; *Slaninová et al., 2014*; *Fan et al., 2015*). Most *Meconopsis* plants are currently threatened because of their low genetic diversity and genetic homogeneity of fixed alleles (*Sulaiman & Babu, 1996*; *Wang et al., 2021*). The cultivation and breeding of *Meconopsis* plants is still in the experimental stage. Elucidating the root microbial composition of *M. integrifolia* and *M. horridula* would provide data that would help support the cultivation and breeding of these Tibetan medicinal plants. A few studies have been published on the chemical composition and pharmacology of *M. horridula* and *M. integrifolia*, but the composition of the root microbiome of *Meconopsis* plants has not yet been studied.

This study used high-throughput sequencing to investigate the microbial community structures of the root zone of two *Meconopsis* plant species, *M. horridula* and *M. integrifolia*, both collected from Tibet, China. This study aimed to (i) investigate whether habitat or plant identity has a more significant impact on the microbial composition of the root zone, (ii) analyze whether there are differences in the composition of fungi and bacteria, and (iii) determine the factors that most impact the microbial composition of the root zone.

## MATERIALS AND METHODS

### Study area and sample collection

Mi La Mountain in Lhasa and the mountain near Dong De Cuo in Nagchu were chosen as study sites. Both *M. horridula* and *M. integrifolia* were collected from the grassy slopes of Mi La Mountain in Lhasa in October 2020, at an altitude of 4,886 m (M site; 29.82°N −92.36°E), and from the mountain near Dong De Cuo in Nagchu, Tibet, China, in July 2020, at an altitude of 4,872 m (D site; 30.99°N−92.94°E). The root zone soil of two plant species and bulk soil were collected from two sites with three replicates, including three root zone soil samples of *M. horridula*, three root zone soil samples of *M. integrifolia*, and three bulk soil samples from the M and D sites, respectively. A total of 18 soil samples were collected (Table 1). Root zone soil is loosely attached to the roots, whereas bulk soil is outside the root zone. For sampling, whole plants were dug up and the soil attached to the roots was shaken into a sterile bag. Bulk soil samples were collected 1 m away from the selected plants. The two species of *Meconopsis* sampled were located in the same habitat, and the replicates were approximately 50 m apart from one another. Each species was also taken within 500 m of another site, ensuring that the two species were obtained from the same habitat. All samples were stored in 95% alcohol at 4 °C for further experiments. The sampling for this scientific expedition was approved by the Forestry Department of the Tibet Region.

**Table 1 Detailed information on the two sample sites.**

**Information of sample sites**

| Sample site | Altitude/m | Longitude (E) | Latitude (N) | Relative humidity (%) | Daytime temperature (°C) | Dew-point temperature (°C) | Illumination intensity (lx) | Samples |
|---|---|---|---|---|---|---|---|---|
| Mountain nearby MILA Mountain (M) | 4,886 | 92.36° | 29.82° | 38.2 | 20.9 | 7 | 1,222,496 | *M. horridula* *M. integrifolia* Bulk soil |
| Mountain nearby DongDe Cuo (D) | 4,872 | 92.94° | 30.99° | 54.5 | 21.8 | 13.4 | 160,064 | *M. horridula* *M. integrifolia* Bulk soil |

## DNA extraction, PCR amplification, and sequencing

The total DNA of the root-soil bacteria and fungi was extracted using the SDS plus enzyme method. To characterize bacterial communities, primers 338F and 806R were used for the amplification of the hypervariable region V3-V4 in 16S rRNA (*Du et al., 2019*). The PCR for the bacterial communities used a 20 μL reaction system that included: 5 × Buffer, 4 μL; 2.5 mM dNTPs, 2 μL; Forward and Reverse Primer (5 μM), 0.8 μL; FastPfu polymerase, 0.4 μL; BSA, 0.2 μL; template, DNA 10 ng; and ddH$_2$O added to a total volume of 20 μL. Premier ITS-1 and ITS-2 were used to characterize the fungal communities (*Gardes & Bruns, 1993*; *Bergelson, Mittelstrass & Horton, 2019*), because they produce fewer non-fungal sequences than ITS3 or ITS4 (*Mello et al., 2011*). The PCR for the fungal communities also used a 20 μL reaction system that included: 10 × Buffer, 2 μL; 2.5 mM dNTPs, 2 μL; Forward and Reverse Primer (5 μM), 0.8 μL; rTaq polymerase, 0.2 μL; BSA, 0.2 μL; template, DNA 10 ng; and ddH$_2$O added to a total volume of 20 μL. The PCR parameters were: 95 °C for 3 min, 95 °C for 30 s, 56 °C for 30 s, and 72 °C for 45 s for 25 cycles; 72 °C for 10 min, then the test was stopped at 10 °C. Nano Drop 2000 was used to test DNA purity and concentration, and agarose gel electrophoresis was used to check for PCR success. All the samples were sent to Shanghai Majorbio Bio-Pharm Technology Co., Ltd., and the sequencing results were obtained through the Illumina HiSeq platform.

## Determination of soil properties

To measure soil properties, available kalium (AK), available phosphorus (AP), organic matter (OM), pH, soil moisture (SM), and total nitrogen (TN) were all measured. Total nitrogen was tested using the modified Kjeldahl Method (Chinese standard method HJ717-2014). Available phosphorus was obtained using the ascorbic acid colorimetric method (Chinese standard method HJ704–2014) (*Ren & Gao, 2022*). Available kalium was estimated using the NY/T889-2004 method, and the NY/T1121.6-2006 method was used for determining soil organic matter (*Lirong, 2019*). These four factors were measured by the Tibet BoYuan Environmental Testing Co., Ltd. Soil moisture was tested with the drying method: 5 g of soil was placed in a drying oven for 1 h and then weighed. The pH was tested using pH equipment (Sartorius PB-10), and the water-soil ratio was 1:1 (*Cao et al., 2022*).

## Statistical analysis

A network analysis was performed on the OTU level using the CYTOSCAPE (CONET) software to estimate the correlation between the fungal and bacterial communities in the root zone. Pearson and Spearman correlations served as the foundation for both positive and negative correlations. A total of 100 randomization interactions were performed and the multiple test correction based on Benjamini-Hochberg was used. When the net sum of the relationships is positive, the impact is synergistic, and when it is negative, the impact of the relationship is antagonistic (*Fath & Patten, 1998*; *Cao et al., 2022*).

An alpha diversity analysis and Student's t test were employed to test the dissimilarity of the structure of the root zone microbiome between plant species and geographical locations. The alpha indexes, including community richness (Chao, coverage and ace) and diversity (Shannon and Simpson), were calculated using a mothur index analysis (version v.1.30.2, https://mothur.org/wiki/calculators/). Then the Shannon index for Student's t test (suitable for a small sample size, $n < 30$) was used to test between-group differences on the class level.

Based on the Bray-Curtis dissimilarity matrix, nonmetric multidimensional scaling (NMDS) was used to explore the similarities and differences in community composition among different grouped samples at the genus level. These grouped samples were then tested through Adonis and the weighted UniFrac distance was calculated with 999 permutations.

A total of five environmental factors were selected based on their *P* value for the canonical correlation analysis (CCA) or redundancy analysis (RDA) on the OTU level. The *P* value, which indicated the relevance of the CCA or RDA, was assessed by ANOVA. CCA and RDA assess the relationship between microbes and measured soil properties. DCA analysis results determined whether an RDA or CCA was used: if the result of the DCA analysis was greater than or equal to 3.5, a CCA was used; if the DCA analysis result was less than 3.5, an RDA was used (*Xiao et al., 2022*).

All the data were analyzed on the free online Majorbio Cloud Platform, except the results of the network analysis. The raw sequence data have been deposited into the NCBI Sequence Read Archive with accession numbers PRJNA814442 and PRJNA813297.

## RESULTS

### Analyses of sequencing data

The MiSeq sequencing analysis of 18 samples resulted in a total of 1,204,304 raw fungal reads and 893,452 raw bacterial reads. The average lengths for fungi and bacteria were 255 and 418 bp, respectively. The dilution curve analysis revealed clear asymptotes, indicating that the fungal and bacterial samples were nearly complete (Fig. 1).

### Microbial composition across two plants

A total of 45 bacterial phyla were detected. The bacterial compositions of *M. horridula* and *M. integrifolia* across the two sites were similar. Actinobacteriota dominated the diversity of the two *Meconopsis* plants from the two different locations, with Proteobacteria following closely thereafter, and Firmicutes, Acidobacteriota, and Chloroflexi were also
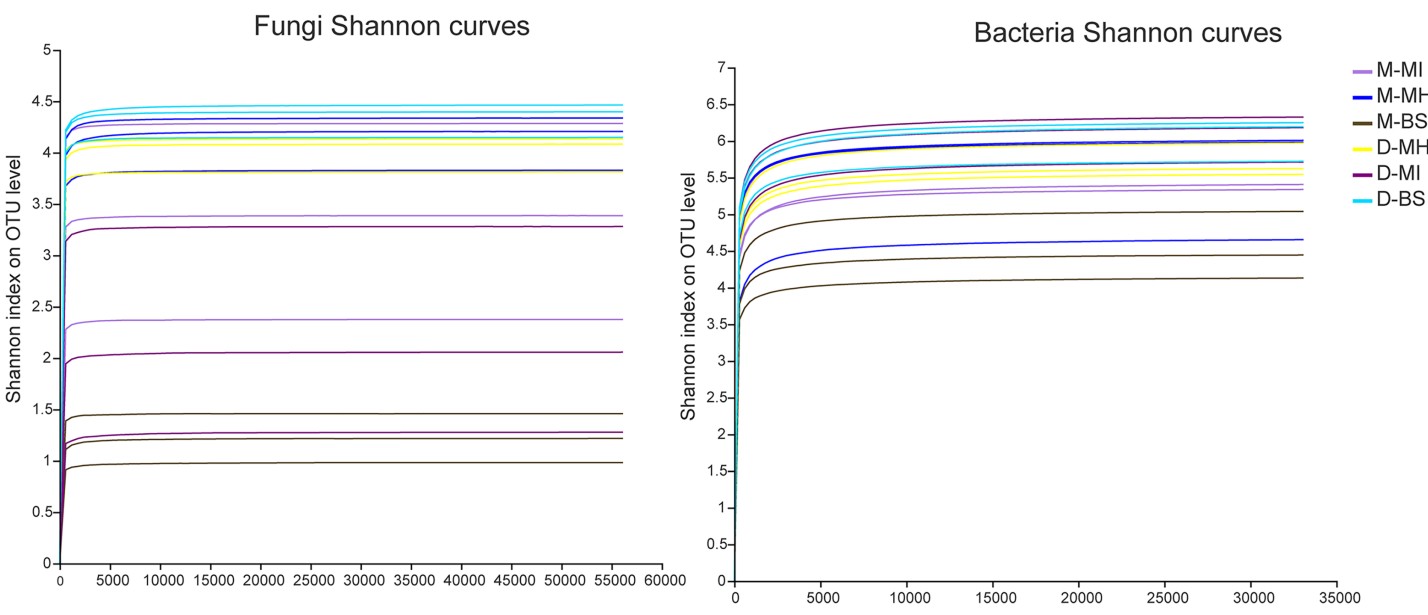

**Figure 1 Shannon dilution curve.** M, Mi La mountain; D, mountain near Dong De Cuo; MH, *Meconopsis horridula*; MI, *Meconopsis integrifolia*; BS, bulk soil.

abundant. The distribution of bacteria across the two plants was similar to that in bulk soil, although the abundance levels were different.

The ITS-1 and ITS-2 sequencing detected a total of 13 fungal phyla. The two plants had different fungal compositions. The phylum Ascomycota was the most numerous in *M. horridula*, accounting for more than half of the OTUs, followed by Basidiomycota and Mortierellomycota. In *M. integrifolia*, Basidiomycota was dominant, accounting for 57.28% of the total fungal composition at site M and 80.37% at site D. Of the fungi detected in the bulk soil, 91.83% were assigned to Basidiomycota at site M, and Ascomycota accounted for 70.24% at site D (Fig. 2).

The top 15 OTUs were selected to estimate the correlation between fungi and bacteria in the root microbiome. The results showed a positive correlation between fungi and bacteria in the root zone. A total of 30 nodes and 178 edges were identified for *M. horridula*, including 68 negative edges and 110 positive edges. For *M. integrifolia*, there were a total of 30 nodes and 296 edges, including 124 negative edges and 172 positive edges (Fig. 3). The synergistic effect was more significant than the antagonistic effect in the correlation of fungi and bacteria.

## Effects of plant identity or habitat on fungi and bacteria

An alpha diversity analysis was used to test the similarity and dissimilarity of root zone microbial diversity to detect whether plant identity or habitat had a greater impact on the root microbial composition of plants on the Qinghai-Tibet Plateau. There were no significant differences observed in the bacterial or fungal communities of the same plant species collected from different habitats. There were also no significant differences in the bacterial diversity of the roots of *M. horridula* and *M. integrifolia* plants collected from the same area. However, there was a considerable difference in fungal Shannon diversity

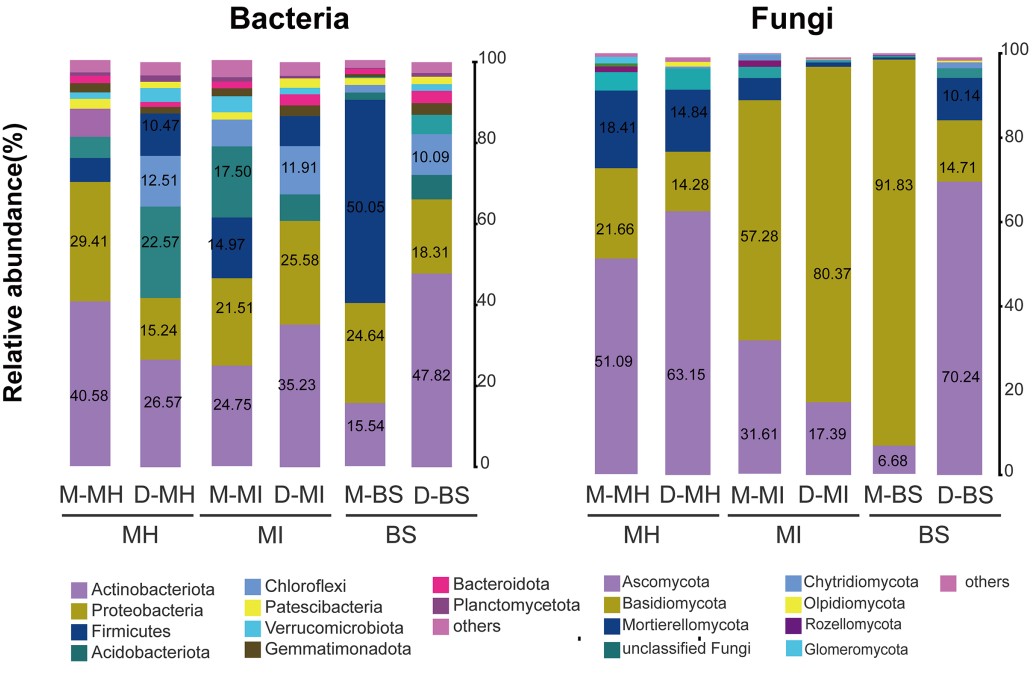

**Figure 2 Relative abundance of fungi and bacteria at the phylum level.** Fungi and bacteria with average relative abundance levels <0.01 were combined into the "Others" category. M, Mi La mountain; D, mountain near Dong De Cuo; MH, *Meconopsis horridula*; MI, *Meconopsis integrifolia*; BS, bulk soil.

between the two plants obtained from the same area. This finding indicates that plant identity has a stronger effect than habitat on fungi and that bacterial diversity was more similar than fungal diversity (Fig. 4).

To further investigate the similarity or dissimilarity of fungal and bacterial communities, an NMDS analysis was performed based on the Bray-Curtis dissimilarity matrix. The NMDS analysis of bacterial communities in *M. horridula* and *M. integrifolia* each revealed a cluster. The *M. horridula* from both locations formed a cluster in fungal communities, although relative plant species from the same habitat did not (Fig. 5). There were more similarities in bacterial diversity and more dissimilarities in the composition of fungi in the root zone of *Meconopsis* species. In summary, the fungal community compositions of root zones were significantly different between the two different plant species, but bacterial structure was similar in the two plants.

## The relationship between microbes and soil properties

Study results (Table 2) showed different correlations between the root zone microbiome and soil properties in *Meconopsis* plants. The *P* value, as determined by ANOVA, demonstrated the relevance of the CCA or RDA. The relevance ranking for bacterial structure was: OM ($P = 0.003$) > SM ($P = 0.004$) > AP ($P = 0.115$) > pH ($P = 0.17$) > TN ($P = 0.376$) > AK ($P = 0.918$). The top five relevant factors, according to *P* value, were then used for the RDA. The relevance ranking for fungal structure was: pH ($P = 0.001$) > TN ($P = 0.001$) >AP ($P = 0.005$) > SM ($P = 0.006$) > OM ($P = 0.043$) > AK ($P = 0.045$). The top

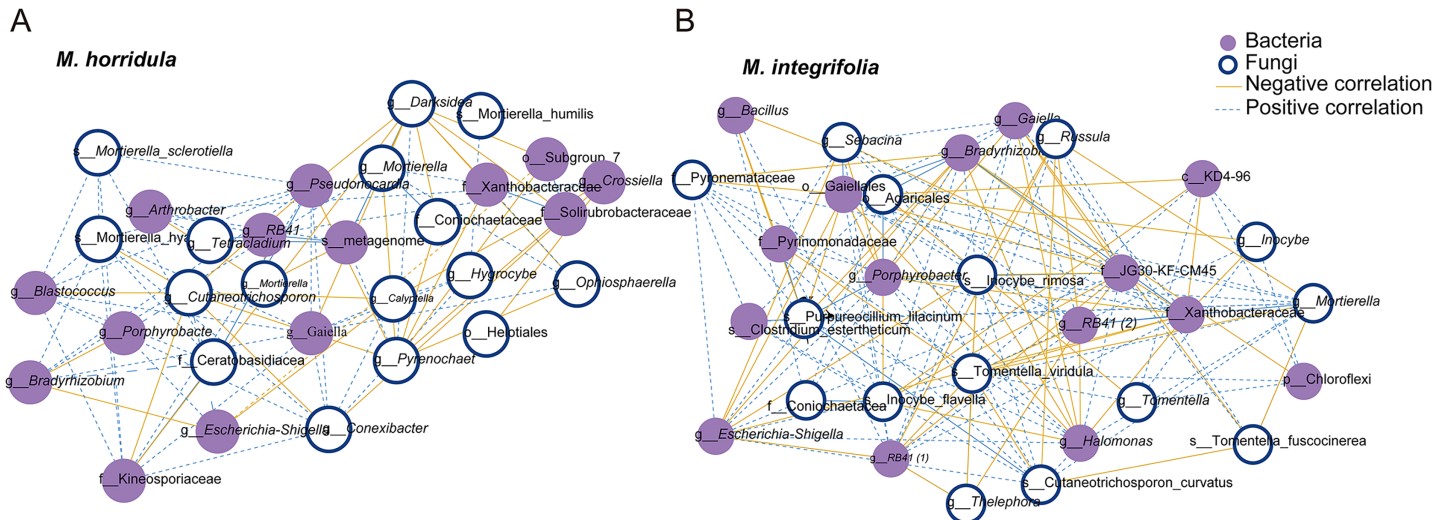

**Figure 3 Network analysis of the correlation between bacteria and fungi.** The top 15 fungal and bacterial OTUs were selected from *M. horridula* (A) and *M. integrifolia* (B). The nodes represent the bacterial (purple) and fungal (blue) OTUs. Edge color represents negative (yellow) and positive (blue) correlations.

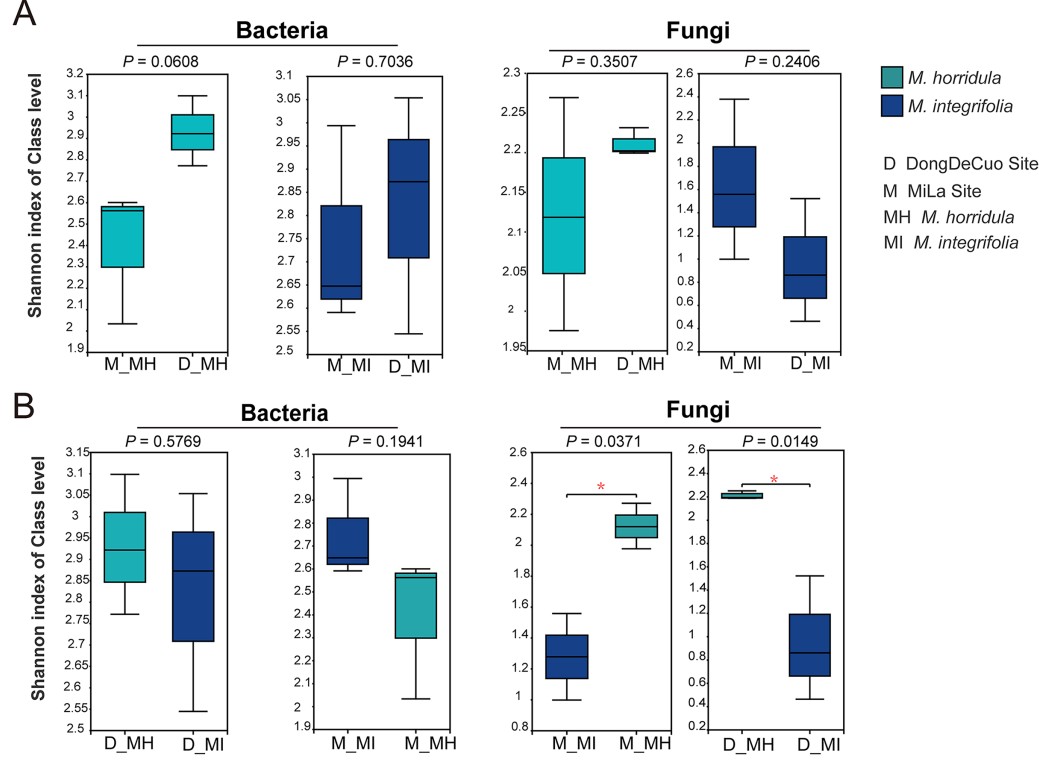

**Figure 4 Alpha diversity analysis of the structure of bacterial or fungal diversity from two sister plants or two geographical locations based on Student's t-test.** Asterisks indicate significant differences (*$P < 0.05$*). (A) The root zone microbiome of plants from different habitats at a class level and (B) Student's t-test analysis for the root zone microbiome of two plants in the same habitat. M, Mi La mountain; D, mountain near Dong De Cuo; MH, *Meconopsis horridula*; MI, *Meconopsis integrifolia*; BS, bulk soil.

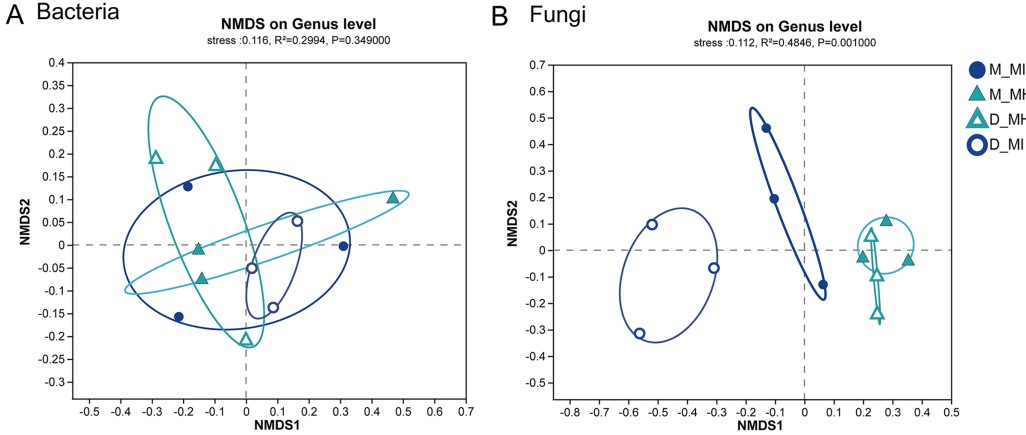

**Figure 5** **Nonmetric multidimensional scaling (NMDS) ordination of root zone bacterial and fungal communities across two host plants from different sites at a genus level.** M, Mi La mountain; D, mountain near Dong De Cuo; MH, *Meconopsis horridula*; MI, *Meconopsis integrifolia*.

**Table 2 Summary of the soil properties of each sample.**

| Site/P value | Sample | Soil moisture (%) | Organic matter/(mg/kg) | Available K/(mg/kg) | Available P/(mg/kg) | Total nitrogen/(mg/kg) | pH |
|---|---|---|---|---|---|---|---|
| M site | *M. horridula* | 13.4 | 42.5 | 18 | 9.6 | 2,430 | 8.13 |
| M site | *M. integrifolia* | 20.9 | 37.6 | 16 | 1.1 | 202 | 6.24 |
| M site | Bulk soil | 36.4 | 112 | 14 | 1.3 | 1,750 | 7.14 |
| D site | *M. horridula* | 27.7 | 66.2 | 14 | 1.5 | 2,880 | 6.94 |
| D site | *M. integrifolia* | 19.4 | 43.7 | 2 | 1.4 | 3,370 | 7.21 |
| D site | Bulk soil | 6.5 | 72.6 | 14 | 4.6 | 3,460 | 7.45 |
| *P* value | (with Bacteria) | 0.004 | 0.003 | 0.918 | 0.115 | 0.376 | 0.17 |
| *P* value | (with Fungi) | 0.006 | 0.043 | 0.045 | 0.005 | 0.001 | 0.001 |

five relevant factors, according to *P* value, were then used for the CCA (Fig. 6). These results demonstrated that soil moisture and organic matter had a major impact on bacterial structure, while pH and available nitrogen were more closely related to fungal structure.

## DISCUSSION

The microbial community composition of the root zones of *Meconopsis* plants from the harsh environment of the QTP is comparable to root microorganisms in low-altitude regions, as shown in previous research. In our study, the structure of fungal communities detected by ITS-1 and ITS-2 in the root zone was influenced significantly by plant identity rather than habitat, but bacterial communities were not significantly affected by either plant identity or habitat. In agreement with numerous studies, the effect of plant identity on the microbiome was significant, especially in fungal communities, which have a

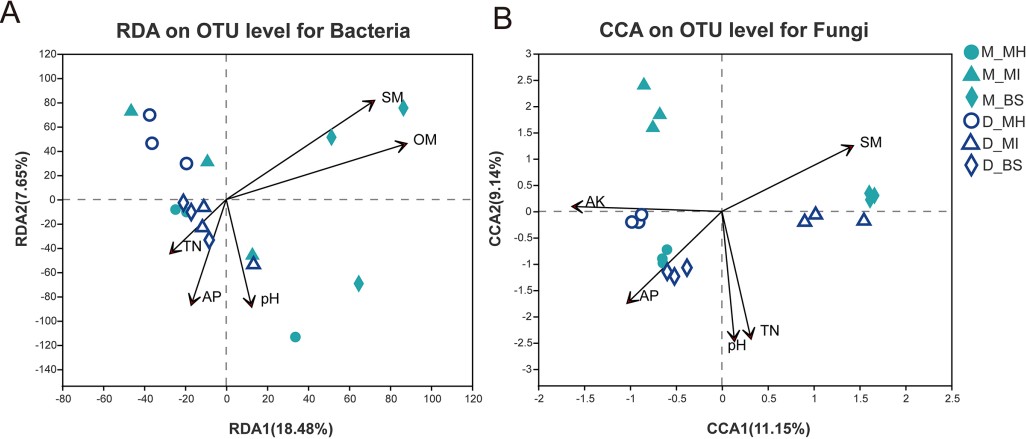

**Figure 6 Redundancy analysis (RDA) for bacterial communities (A), Correspondence analysis (CCA) for fungal communities (B), and their interaction with soil properties.** M, Mi La mountain; D, mountain near Dong De Cuo; MH, *Meconopsis horridula*; MI, *Meconopsis integrifolia*; BS, bulk soil.

stronger correlation with plant identity in contrast to other microbial communities (*Millard & Singh, 2010*; *Zinger et al., 2011*). Plant-fungal communities have a stronger impact on plant identity than plant-bacterial correlations (*Bergelson, Mittelstrass & Horton, 2019*). As the climate gets warmer on the QTP, the dissimilarity of root fungal communities between different plant species may increase due to potential changes in root exudates or disruptions in the adaptability of plants due to active selection by the host plant (*Jiang et al., 2021*). These findings suggest that fungi may be important to maintaining the stability of microbial communities.

There were significant differences in the root zone fungal composition of the two plant species, whereas bacterial composition was similar in the two plant species between the two sites. Two factors may explain this finding. First, fungi may significantly improve plant pathogens and environmental tolerance, particularly in harsh environments. This has been demonstrated in some studies of endophytes (*Rodriguez et al., 2008*; *Rodriguez, Woodward & Redman, 2010*). The root zone microbiome may play the same role as microbiota, where plants can communicate and recognize each other through plant-specialized metabolism, allowing the plants to modulate root-associated microbiota based on the plant's needs. In *Arabidopsis thaliana* (L.) Heynh, for example, triterpene compounds selectively affect root microbiome assembly (*Huang et al., 2019*). Second, the similarity of bacterial communities in roots allows plants to quickly colonize new habitats, improving their fitness. Fungi and bacteria play different roles in plant growth and stress tolerance. Previous studies have focused on the interaction between endophytic fungi and plants. The symbiosis of fungi as endophytes improves the stress tolerance of the environment and pathogens (*Márquez et al., 2007*; *Rodriguez et al., 2008*; *Rodriguez, Woodward & Redman, 2010*; *Fitzpatrick et al., 2018*). Some components of the root zone microbiome, such as ectomycorrhizal fungi, benefit plant growth (*Kumar & Atri, 2018*; *Guerrero-Galán, Calvo-Polanco & Zimmermann, 2019*, *Aryal, Meiners & Carlsward, 2021*), but the functions of other members of these microbial communities remain unclear. Investigating

the influence of root zone microbial communities enables the analysis of the impacts of plant identity and the geographical environment in manipulating the root-associated microbiome.

In contrast to our findings that fungal communities were impacted by plant identity, but not habitat, *Yao et al. (2013)* discovered that fungal communities in the root were influenced significantly by habitat. The composition of root-associated fungal communities differed across sampling sites, and these differences were linked to soil nutrient concentrations. Numerous studies have found that soil properties can influence microbial communities, but the impact varies depending on the habitat (*Li & Liu, 2019*; *Zhang et al., 2019*; *Wu et al., 2021*).

Total nitrogen and pH were the most significant factors affecting the structure of fungal communities in our study, whereas soil moisture and organic matter had the most significant impact on the structure of bacterial communities. The relationship between fungal richness and nitrogen concentration is strong (*Jiang et al., 2021*). In contrast to our findings, a previous study found that pH was positively related to the relative abundance and diversity of bacteria, but less related to fungal communities (*Cao et al., 2020*; *Rousk et al., 2010*). All of the samples in our study had alkaline pH values, and the pH effects may not be apparent with pH value changes as narrow as they were in our study, which may explain these differing results (*Fierer, 2017*).

The environment has been shown to impact the influence of soil properties on microbial communities. Moisture was more related to microbial composition in the warm dry sites and cool wet sites (*Brockett, Prescott & Grayston, 2012*). The impact of the interaction of soil properties on microbiome diversity and structure differed from the effect of a single factor (*He et al., 2016*). The interaction of factors, soil type, and nutrient conditions should all be considered when discussing the relationship between soil properties and the plant microbiome. On the QTP, there is currently no fixed index for evaluating soil nutrition. Due to differences in soil types and soil nutrition, it is impossible to determine which factors have a greater impact on microbial structure. More controlled experiments are required to further study the significance of environmental factors on plant-microbe interactions in this area.

Plants can function as ecosystem engineers, shaping the microbiome associated with their roots to their advantage (*Coyte, Schluter & Foster, 2015*). Co-occurrence networks can be used to study microbe–microbe interactions and to explain microbial community stability (*Barberán et al., 2012*; *de Vries et al., 2018*). The interaction of bacterial and fungal communities has an impact on the stability of these communities in harsh environments (*Coyte, Schluter & Foster, 2015*; *Jiao et al., 2022*). Negative feedback in microbial communities restrains positive feedback networks, which benefits the host plant and increases stability (*Coyte, Schluter & Foster, 2015*). However, in our study, the impact of positive feedback was stronger than the impact of negative feedback in the correlation of fungal and bacterial communities in the root zone microbiome of *Meconopsis* plants. This could be because of alpine habitat heterogeneity. Positive or negative interactions between

bacterial and fungal communities are a tradeoff, and synergistic interactions could be beneficial in some cases (*Mille-Lindblom, Fischer & Tranvik, 2006*).

In this study, Basidiomycota and Ascomycota were the dominant fungi in the root fungal communities, according to the ITS-1 and ITS2 sequencing results. Even though the ITS-1 and ITS-2 primers we used for fungi could offer adequate information for taxonomic assignment and are commonly used in amplicon pyrosequencing studies of fungal diversity (*Mello et al., 2011*; *Bergelson, Mittelstrass & Horton, 2019*), it is still unclear if they provide the best taxonomic resolution of the species (*Mello et al., 2011*). More ITS regions should be used in order to increase classification accuracy in future studies. The 16S rRNA results showed that Actinobacteriota, Proteobacteria, and Acidobacteriota were the dominant bacteria in the root bacterial communities of *Meconopsis* plants. Among the microbial communities we detected, the phyla Actinobacteriota and Proteobacteria were also the dominant bacteria of radiation-tolerant bacteria isolated from the soil in Tibet (*Rao et al., 2016*). In high alpine permafrost, Proteobacteria dominate the bacterial communities. To adapt to the environment, they minimize anaerobic metabolism and live ectosymbiotic lives (*Frey et al., 2016*). This could explain the predominance of Proteobacteria in the root zone of *Meconopsis* plants on the QTP. Certain bacteria may have a long-term association with plants (*Yeoh et al., 2016*). The phyla Basidiomycota and Ascomycota were found to be dominant in our study and are also the dominant fungi of *A. thaliana* plants (*Bergelson, Mittelstrass & Horton, 2019*). In other studies, the fungus Ascomycota is distributed over a large range of scales, and it is the dominant fungal phylum in Tibet (*Maestre et al., 2015*; *Prober et al., 2015*; *Yang et al., 2017*). A better understanding of the root microbial diversity and assembly characteristics of wild Tibetan medicinal plants could improve cultivation efforts of these plants and add to the knowledge of root zone soil microorganisms on the Qinghai-Tibet Plateau.

## CONCLUSIONS

This study revealed that plant identity had a larger impact than habitat on the root zone fungal communities of two Tibetan medicinal plants, and that root bacterial communities did not differ between the two plant species or between the two sites. These findings demonstrate that the root zone microbial composition of fungi and bacteria respond differently to plant identity and habitat. We speculate that root fungal communities have a greater impact on plants than bacterial communities and that the fungi-plant relationship should be an area of future research. Furthermore, in the connection of fungi and bacteria, the synergistic effect was more significant than the antagonistic effect. Fungal community structures detected by ITS-1 and ITS-2 were influenced by available nitrogen and pH, whereas bacterial structures were significantly influenced by soil moisture and organic matter.

## ACKNOWLEDGEMENTS

We acknowledge BoJie Lu (South-Central Minzu University) for his help with statistical analyses.

## Funding

This work described in this manuscript was supported by the Local Development Funds of Science and Technology Department of Tibet (Nos. XZ202001YD0028C; XZ202102YD0031C); the Wuhan University Plateau Ecology Youth Innovation Team Fund (413100105); and the Tibet Autonomous Region Science and Technology Project (XZ202001ZR0023G); and Tibet Maidika Wetland Construction Project (Tibet Financial Advance Indicator [2023] No. 1). The funders had no role in study design, data collection and analysis, decision to publish, or preparation of the manuscript.

## Grant Disclosures

The following grant information was disclosed by the authors:
Local Development Funds of Science and Technology Department: XZ202001YD0028C; XZ202102YD0031C.
Wuhan University Plateau Ecology Youth Innovation Team Fund: 413100105.
Tibet Autonomous Region Science and Technology Project: XZ202001ZR0023G.
Tibet Maidika Wetland Construction Project: Tibet Financial Advance Indicator [2023] No. 1.

## Competing Interests

The authors declare that they have no competing interests.

## Author Contributions

- Shuting Chen conceived and designed the experiments, performed the experiments, analyzed the data, prepared figures and/or tables, authored or reviewed drafts of the article, and approved the final draft.
- Pengxi Cao analyzed the data, prepared figures and/or tables, authored or reviewed drafts of the article, and approved the final draft.
- Ting Li analyzed the data, authored or reviewed drafts of the article, and approved the final draft.
- Yuyan Wang performed the experiments, prepared figures and/or tables, and approved the final draft.
- Xing Liu conceived and designed the experiments, authored or reviewed drafts of the article, and approved the final draft.

## Data Availability

The raw sequence data are available at the NCBI Sequence Read Archive: PRJNA814442 and PRJNA813297.

## Supplemental Information

Supplemental information for this article can be found online at http://dx.doi.org/10.7717/peerj.15361#supplemental-information.

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
