# Peer review of "Microbial diversity patterns in the root zone of two Meconopsis plants on the Qinghai-Tibet Plateau"

_PeerJ, doi:10.7717/peerj.15361_

## Round 0.1 · original submission · Major Revisions

Please carefully address the comments raised by Reviewer 1 and clarify the statement in the text as suggested by Reviewer 2.

Also, please clarify the limitations of techniques and kinds of sample the authors used, and make conclusions based on the evidence.

·

Basic reporting

The authors present an interesting study on root microbiomes of medicinal plants which happen to be sister species. In principle the study is straight forward, but some methods need to be clarified. Moreover, the authors make way too general conclusions out of a very limited experimental design. Thus the conclusions in the abstract and the conclusion section should be tone down, and the limitations should be deeply discussed in the discussion section.

Experimental design

Below there are some other specific comments:

1. Authors should state clearly how many plants of each species they studied. From the total number of samples it is not easy to figure out how many individual plants per species were studied.
2. The primers used for fungal communities have biases towards certain taxa. They should state this in the text, discuss it in the results and state it again in the discussion. Otherwise the conclusions are biased.
3. The use of nanodrop to quantify DNA concentration and purity is far from the gold standard: bioanalyzer or qbit.
4. The methods nor the results specify how taxonomic categories were assigned. Explain why to use OTUs, rather than ASVs.
5. On the correlation Table 2, it is hard to see how out of two sample sites, and two species, it is possible to obtain biological meaningful correlations. By the way, Table 2 was missing.
6. In general, I feel like it is hard to draw the type of conclusions that the authors attempt (the importance of host specificity) on bacterial and fungal communities, from two sites and two sister species. The authors should tone down the discussion and conclusions, as at the moment it is only relevant to to site species from two sites.
7. The raw sequence data should be made available in a public repository.

Validity of the findings

The conclusions are way too ambitious given the experimental design. They should be tone-down.

·

Basic reporting

1. Basic Reporting

This interesting study describes the bacterial and fungal communities associated with the root zones of Meconopsis plants in the Qinghai-Tibet Plateau, suggesting that the fungi associated with the plants tended to be more host specific than the bacteria.

Clear and unambiguous, professional English used throughout.
Yes, this manuscript is written in clear professional English throughout. In a few spots, I was confused by the words used, but this didn’t happen often. (I have highlighted these instances below.)

Minor points of language confusion:
L20: what are “fitness microbial features”? I’d recommend rephrasing this if possible.
L30: what is the “abundance of fungal communities”? Do you mean that the relative abundance of fungal species was influenced by total N and pH? Or did total fungal abundance change? I’d recommend rephrasing to clarify this.
L41: what is “dry pressure”? Is this negative water potentials? Or just drought stress? Again, rephrasing might help.
L68: What is a “PhytOakmeter”? Please make this clear when you mention it in the manuscript.
L155-158: “Selection principle of RDA or CCA model are based on DCA analysis… choosing RDA are better than CCA” These sentences read less clearly than the other portions of the manuscript. If possible, it would be great to rephrase this to see if you can make it a bit clearer.

Literature references, sufficient field background/context provided.
This manuscript cites many relevant articles, and I commend the authors for reading as thoroughly as they have. However, I think the paper would be even stronger if they were to cite more broadly, or provide more context and information about the ideas they explore.

For instance, in the introduction (L64-66), the authors assert that “previous research on species specificity or habitat specialization of microbial communities focused on endophytes and rhizosphere communities,” and go on to imply that “root zone” soil is often overlooked. This may be true, but 1) how do you define the root zone as distinct from the rhizosphere? Or are these the same?, and 2) I think the root zone, if by this we mean a slightly larger soil volume than a typical “rhizosphere,” may not be super neglected (which is good, it puts your project into a broad context!). A lot of studies have sequenced soil in the vicinity of roots to look for microbial patterns related to host association and habitat specialization (certainly this happens a lot in mycorrhizal literature); it might be worth citing some of this literature here.

Professional article structure, figures, tables. Raw data shared.
This article is professionally structured with nice figures and tables. Raw data are shared as supplementary information (soil chemistry) and through the NCBI SRA (accession numbers provided). Minor comments below.

1. Figure 3: “bacterial (blue)…” should probably read “bacterial (green).” At least, the bacterial nodes look green on my screen!
2. Figure 5: Which panel is which? (I assume left is bacteria and right is fungi, but please clarify in the legend.) Why is the fungal plot missing any dots for D-MH? Please explain this omission in methods or results, and ideally acknowledge it briefly in the figure legend so readers aren’t confused.
3. Table 1: What statistical test are these P values from? Please specify somewhere in the manuscript – perhaps cite Table 1 after the place in the methods where you describe the test you did to produce these P values.

Self-contained with relevant results to hypotheses.
Yes, this is a self-contained manuscript whose results address the questions posed.

Experimental design

2. Experimental design
Original primary research within Aims and Scope of the journal.
Yes, this manuscript fulfills this criterion.

Research question well defined, relevant & meaningful. It is stated how research fills an identified knowledge gap.
Yes, this paper sets out to describe the microbial communities associated with two Meconopsis species, and explore the factors structuring these communities. This seems to be a knowledge gap that could be important for better understanding the biology of this important medicinal species.

Rigorous investigation performed to a high technical & ethical standard.
Yes, the methods seem scientifically appropriate and to have been ethically done.

Methods described with sufficient detail & information to replicate.
Yes, this methods section provides nearly all the information I would expect to see in a published manuscript.

One change I would request: I would like to see the network analysis described in a few sentences (currently just described as “the default process in the Microbial association network construction tutorial”), rather than relying on a citation of a website that could become inaccessible in the future (L142-143).

Minor questions:
L124: what is “completeness of DNA fragments”? I have used agarose gels to check for PCR success in the past (i.e. single, strong bands). Is that what you mean?
L136: should “dying” read “drying,” instead?
L150: Were these Bray-Curtis dissimilarities calculated at the OTU level, or higher level taxonomy? Your figures show different levels of organization at different times, so I’d love to know at what taxonomic level these calculations were performed.

Validity of the findings

3. Validity of findings
Impact and novelty not assessed. Meaningful replication encouraged where rationale & benefit to literature is clearly stated.

All underlying data have been provided; they are robust, statistically sound, & controlled.
Yes, this appears to be the case.

Conclusions are well stated, linked to original research question and limited to supporting results
In general, I believe this to be true. I think the authors could conduct some simple additional analyses (described in “general comments”) to strengthen their claims, if desired, but the work they have already done is also publishable and interesting.

Minor comments:
L237: You say that “several factors” might explain the different host specialization patterns between fungi and bacteria, but then only name two (1 – host plants might control fungi based on their needs; 2 – having similar root bacteria could help colonize new habitats). Superficially, you could change the word “several” to “two” and be done correcting the phrase. However, don’t these factors kind of contradict each other? Does the plant manipulate its microbiome to be maximally useful to itself (in which case you should see host specialization), or does it maintain a broad portfolio of partners in order to colonize diverse new habitats? If it does the first for fungi, and the second for bacteria, why should these two types of microbes get such different from the plant? Might be nice to see a sentence or two diving a little deeper into this.
L293: “The fungal and bacterial compositions of related species in Meconopsis plants were similar to those in bulk soil at the phylum level, illustrating that the root microbiome was recruited from the surrounding bulk soil…” It may be true that the root zone microbes are recruited from the surrounding soil, but asserting this on the basis of an analysis at the phylum level seems like a stretch to me. What does it look like at the OTU or ASV level?

Additional comments

4. General comments

I enjoyed reading this article and thank the authors for their commendable work on this study. I had a couple of questions whose answers might also be of interest to future readers of this article, and so might be worth addressing in the revised submission:

1) What was the surrounding vegetation like at the sites where you sampled M. horridula and M. integrifolia? Your network plots (Fig. 3) include kind of a lot of likely ectomycorrhizal species (e.g. Tomentella viridula, Inocybe rimosa, etc.). Why do you think this is? Were there ectomycorrhizal host plants nearby? Is Meconopsis hosting these fungi? Some of them seem to be quite dominant in the root zone of the plants!
2) Do your conclusions change if you conduct your analyses consistently at an OTU or ASV level of taxonomy for the fungi and bacteria? I was surprised to see so much phylum-level analysis in this paper, since it’s very hard to infer ecological function (like host specialization) from phylum-level classifications.
3) Did you consider performing a PERMANOVA (R, vegan::adonis() ) and beta dispersion analysis on your Bray-Curtis dissimilarity matrices? This would be a typical way to examine the impact of variables like host and habitat on microbial community structure, so I was kind of surprised not to see it here, although the ANOSIM probably achieves more or less the same thing here. It would also be nice to see an indicator species analysis, or a species classification method test (vegan::clamtest() ) to explicitly test for specialized microbial taxa on these hosts, to strengthen the claim that the bacteria and fungi had such different patterns of specialization. However, these additional analyses aren’t strictly necessary to have a useful and interesting manuscript, if you are satisfied that you have already adequately described the system.

---

## Round 0.2 · Minor Revisions

- Please address the comments precisely with relevant literature.
- Also, discuss and conclude your findings based on your evidence.
- If there are some limitations in terms of techniques used, please clearly explain.
- I understand that the plant-microbe interaction in the environment is a complex process, and several factors affect their living and diversity. I think the authors should suggest some ideas (e.g., techniques used to provide stronger evidence) for further study to increase reliability and reduce bias.

·

Basic reporting

The authors improved the manuscript but there are still methodological issues that should be clearly stated.

Experimental design

Perhaps the research is not perfectly designed, mainly because of economic limitations, thus it has a low number of samples. In this regard, I am still not satisfied with the response of the authors, and the addition of one sentence to the main text. It should be edited so the explanation is crystal clear. Also the wording of headers in Table 1 should be edited. It reads "Sample (amount)", This is not an appropiate way to refer to the number of replicates.

As for the biases given by the ITS regions used, it is not enough just to cite a paper from 2011. Yes, ITS has been used in the field, but the field is also moving on real fast. And it has been acknowledged that there are biases when using few ITS regions. This is not trivial, as the authors draw conclusions about the relationship between taxonomic units, species and the environments. These issues need to be further explained in the methods, resuts and discussion.

Validity of the findings

The conclusions were toned down, but perhaps they need to be further toned down given the biases that the studied ITS regions have.

·

Basic reporting

This remains an interesting study describes the bacterial and fungal communities associated with the root zones of Meconopsis plants in the Qinghai-Tibet Plateau, showing that the fungi associated with the plants tended to be more host specific than the bacteria. The authors have improved the manuscript quite a bit since their original submission.

Language
Yes, this manuscript is written in clear professional English throughout, with occasional remaining exceptions. (I have highlighted these below.)

L163-166 (model selection, "Selection principle of RDA..."): I appreciate the authors' adding some extra detail here, at my request, but I think perhaps the language is getting in the way of my understanding this section completely. I suspect this could be cleared up by a skilled copy-editor (I think a few well placed articles would clear things up?), but as it is, I had trouble following what this selection process is like, since I've never done it myself.
L278: "the pH was not related to bacteria when the pH values were in a narrower range..." What do you mean by this? Perhaps you're saying that the pH would not have been a strong predictor of bacterial diversity if the soils had been less alkaline, or if they had exhibited a smaller range of pH values? Please clarify in the text.


References
This manuscript cites many relevant articles. I would ask the authors to include just a little bit more mycorrhiza literature when considering their root zone findings (as noted below), and I was puzzled by one reference to a paper about Yersinia pestis in mammalian lungs. Otherwise the references seem okay.

L262: I appreciate that the authors added citations about endophytic effects on plant health, but I remain convinced that it is misleading to say that "whether the root zone microbiome also has beneficial effects on plant growth, as endophytic fungi do, still needs further investigation," without going on to cite some of the many, many papers that have sequenced root zone fungi and found links to plant health. Crucially, I would assert that mycorrhizal fungi will be found abundantly in the "root zone," and have well documented positive effects on plant health. (I suspect you can also find pathogens, e.g. Armillaria, in the root zone, with clear negative consequences for plant health.) Many studies of these mycorrhizal fungi in natural ecosystems characterize communities by sequencing the soil near roots (but not rhizosphere), i.e. the root zone. Please engage at least a little bit with this literature. I think I would feel a lot more comfortable with this section if you just added a sentence or two to acknowledge this, e.g. "It is likely that some members of the root zone microbiome, such as mycorrhizal fungi, have beneficial effects on plant growth (citations), but the functions of other members of these microbial communities remain unclear."
L433: This paper is about Yersinia pestis in the lung, surely this is not what you meant to cite when discussing fungal endophytes? Please remove or clarify why the mammalian disease paper is in here; worth double checking the other citations as well to make sure they are what you want.

Structure and data
This article is professionally structured with nice figures and tables. Raw data are shared as supplementary information (soil chemistry) and through the NCBI SRA (accession numbers provided).

Self-contained
Yes, this is a self-contained manuscript whose results address the questions posed.

Experimental design

I have nothing to add to my prior review, which the authors have responded to satisfactorily. The authors have done a nice job in this area.

Validity of the findings

Once again, I have nothing to add beyond what I have already said. The authors’ responses have satisfied any concerns I raised.

Additional comments

Additional minor points:
L195-197: This is the first time after the abstract that you mention "synergistic" and "antagonistic" effects from your network analysis. What do you mean by this? If two nodes were positively correlated, and so had a positive edge between them, was that considered to be a "synergistic" effect, while a negative correlation/edge would be "antagonistic"? Please explain briefly in the methods.
L258: "Choose certain fungi over bacteria because fungi are more closely related to plants (Bergelson et al. 2019)" This statement does not seem correct to me, and is not supported by the citation given, as far as I can tell. I have never heard it asserted before that plants are better able to filter fungal partners because they share a more recent common ancestor. Probably best to remove this statement, or explain further.

---

## Round 0.3 · Minor Revisions

The scientific content is much improved. However, English should be proved by a proficient English speaker, or editing service, before the manuscript can be accepted for publication.

---

## Round 0.4 · accepted · Accept

The current version is much improved regarding scientific content and English.
After carefully reading it, I am pleased with this version, and it is ready for publication.

Congratulations!!!